# MHC Class I Regulation: The Origin Perspective

**DOI:** 10.3390/cancers12051155

**Published:** 2020-05-04

**Authors:** Alicja Sznarkowska, Sara Mikac, Magdalena Pilch

**Affiliations:** International Center for Cancer Vaccine Science, University of Gdansk, 80-308 Gdansk, Poland; sara.mikac@phdstud.ug.edu.pl (S.M.); s1797808@sms.ed.ac.uk (M.P.)

**Keywords:** viruses in evolution, ERVs, HERVs, adaptive immunity, MHC class I, non-coding RNAs, miRNAs

## Abstract

Viral-derived elements and non-coding RNAs that build up “junk DNA” allow for flexible and context-dependent gene expression. They are extremely dense in the MHC region, accounting for flexible expression of the MHC I, II, and III genes and adjusting the level of immune response to the environmental stimuli. This review brings forward the viral-mediated aspects of the origin and evolution of adaptive immunity and aims to link this perspective with the MHC class I regulation. The complex regulatory network behind MHC expression is largely controlled by virus-derived elements, both as binding sites for immune transcription factors and as sources of regulatory non-coding RNAs. These regulatory RNAs are imbalanced in cancer and associate with different tumor types, making them promising targets for diagnostic and therapeutic interventions.

## 1. Introduction

Viruses are the most numerous and diverse agents in every habitat so far examined, and species-specific virus-derived information make up a substantial component of all organisms’ genomes and epigenomes [1,2,3,4,5]. Over the years, endogenized retroviruses (ERVs) and their defectives have been shown to actively shape the structure and regulation of basically all species’ genomes [6,7,8,9]. The discovery of the involvement of ERVs’ envelope proteins (syncytins) in the trophoblast formation during placentation has shaped the view on a virus–host relationship and has led to the adoption of a perspective that viruses have repeatedly provided useful genes that have been ‘co-opted’ or ‘exapted’ by the host [10,11,12,13,14]. Later, with the emergence of high-throughput sequencing techniques, it became apparent that viral defective elements (which do not encode any functional proteins) largely outnumber the ERV-encoded open reading frames (ORFs) and function as promoters, enhancers, and sources of regulatory RNAs [15,16,17,18,19,20]. Still the currently accepted perspective is that ERVs and their defectives are the remnants of past viral ‘plague sweeps’ retained in the genome of the surviving host, who cunningly co-opted them for structural and regulatory functions. This is the ‘host comes first’ perspective, which implies that point mutations and selection of the fittest type are sufficient to edit pre-existing code and assign it a new meaning (co-opt genes or regulators). In this work, we present an alternative ‘virus first’ perspective proposed by Luis Villareal and Gunther Witzany, where persistent populations of viruses are the agents competent in using and editing the code of their host, thus constituting a driving force of the host evolution. Persistent infection by a virus population alters the genetic background of the host (identity) and strengthens its immunity. Indeed, even the earliest virus–host studies proposed that viral infection protects against similar viruses. Here, we bring forward viral origins of adaptive immunity and review the current knowledge on MHC class I regulation by virus-derived elements. Finally, we seek for links between the aberrant expression of these elements and cancer occurrence.

## 2. Living in the Virosphere

With the onset of large-scale sequencing techniques, it has become evident that viruses are the most prevalent entities on the planet, present in all habitats. They infect each and every living species and we exist in a ‘virosphere’, a cloud of diverse viral populations, where genetic information is constantly changing between viruses and their hosts [21,22] (Figure 1). In the genomes of all living beings, most prevalently in non-coding regions, we find remnants of multiple viral colonization events, usually in the form of solitary long terminal repeats (solo LTRs) [22,23]. These genomic viral elements were initially believed to impair the proper expression of genetic material and contribute to diseases. This notion has been re-evaluated and based on extensive prokaryotic studies, it has been instead proposed that viruses are responsible for horizontal gene transfer, not as simple gene vehicles, but as sources of new genes and their regulators [24,25]. A growing number of reports indicate that viral elements account for a regulatory capacity that stands behind the complexity of the Eukaryotic genome [15,16,17,18]. The 20,000 protein coding genes in the 1000-cells worm *Caenorhabditis elegans* is similar to that of humans. Although most of these genes encode proteins of similar function, the differences in the complexity between the species are enormous, so it became apparent that the regulatory complexity stands behind the developmental complexity. Regulatory non-coding RNAs (ncRNAs) mainly originate from parasitic “junk DNA” sequences, and complex regulatory networks like the immune or nervous systems are especially rich in regulatory elements derived from viruses [20,26,27]. According to Luis Villarreal, the virus-driven exchange of genetic information allows the host to survive in its virosphere and requires a persistent virus–host relationship [22,28]. Persistence is a viral strategy to maintain a continuous presence in its host and transmit to next generations. How the persistence is acquired is explained by the means of an “addiction module”, a strategy in which the virus is bound to the host cell by making the cell addicted to the virus persistence. It was postulated that this strategy represents a general mechanism of binding two linages of life in order to originate symbiosis [22].

## 3. Survival in the Virosphere Thanks to Viral Addiction Modules

Among retrovirus-derived elements in the human genome, there are around 330,000 solo LTRs (retroviral long terminal repeats) that have mostly originated from full retrovirus insertions [22]. These LTRs provide complex gene regulation in networks like the interferon response [29], primate p53 network [30], and placenta development [14]. The prevailing view is that viral elements are the evolutionary remnants of past viral infections that have been ‘exapted’ or ‘co-opted’ by the host for new genes and regulatory elements (promoters, enhancers, gene silencing), though an alternative explanation has also been suggested [22,31], based on the mechanism by which the colonizing viruses acquire persistence in their host’s genome. This strategy is known as an addiction module in which “the virus has addicted the host to its presence and created the new virus–host entity that is more successful in the virosphere” [22].

The addiction module mechanism was first described by Yarmolinsky and colleagues in 1993 during their studies on the P1 phage infecting *Escherichia coli* [32]. They were trying to understand how the episome form of P1 could stably infect the bacteria and why the cell dies when the phage is lost, and found that P1 was using an addiction strategy to promote its retention in the cell. P1 genome encodes for a stable toxin (T) and a less stable antitoxin (A). If P1 episome is lost, the more stable toxin would kill the host. The counteracting gene functions of toxin and antitoxin are the exemplar of how the virus can make a cell “addicted” to its presence [22,31]. It has been further concluded that an addiction module allows for merging two genetic lineages into one, as the P1 phage and *E. coli* have now become one entity, able to oppose other lytic viruses from its virosphere (such as T4 lambda). Infection by T4 lambda disrupts the addiction module, releasing the killing effect of a toxin, and results in the death of an infected cell. This can be seen as a form of immunity in which the P1 infected cell becomes ‘immune’ to similar lytic viruses. The new virus–host entity then provides a survival advantage, especially in the virus-rich environment. Due to the transmissive nature of the virus, the survival applies to groups that were virus-colonized. Thus the combined killing–antikilling effect of the virus defines and provides a group identity (linked to immunity) [27,31].

The persistence of viruses in Eukaryotes cannot be explained by the P1 toxin/antitoxin system as eukaryotic viruses do not typically encode for toxins and antitoxins. Here, the toxic (lethal) effect is directly derived from the virus capacity to induce cell lysis, balanced by the virus persistence, which is cell protective and confers immunity. However, there is perhaps something to learn from the phage concept in that viral persistence protects the cell from an infection of the same or similar lytic viruses. Indeed, it was shown that human endogenous viruses protect against exogenous virus infections [9]. An interesting question is how the virus confers its persistence in the eukaryotic cells where there has been a shift towards viruses that integrate their information into the host cell’s DNA. While DNA viruses and plasmids predominate in prokaryotes, eukaryote genomes mainly host DNA derived from retroviruses, rcr-DNA viruses, retroposons, and transposons. Since the majority of these agents code for non-protein-coding RNAs, it reflects a big change in the regulatory strategy. While the amount of protein encoding genes has remained relatively static, a number of these parasitic viruses and agents have increased rapidly during the eukaryote evolution [26]. A majority of cellular processes is controlled in various aspects by non-coding RNA species and, thus, viral integration in the non-coding sequences has played a role in shaping the function of these RNAs [22]. In line with this, it has become evident that eukaryotic viruses encode miRNAs, establishing their persistence by various ways in the host genome [33,34,35,36,37]. It has been suggested that viruses orchestrate both host and viral ncRNA regulation to balance replication and infectious state (for example, latent versus lytic infection). miRNA regulation can be reversed upon certain cellular stresses, indicating that the addiction module can be disrupted when sufficient viral information integrity is lost [36].

Survival in the virosphere is therefore promoted by the enhancement of immunity resulting from virus persistence. In this model, any new successful viral colonization needs to be consistent with an already operating addiction module, thus it will also promote the network modification. Taking this perspective, any programmed cell death mechanism should be involved in immune response and self-identification and is likely to have been derived from persistent viral colonization events. The complexity of these mechanisms indicates the complexity of regulatory networks originating and enhanced by persistent viruses.

The ability of viruses to create regulatory networks was suggested to result from their competency in using and editing the genetic code [38]. This conclusion was made when applying the laws of linguistics to molecular genetics. If DNA is considered as a natural code or a language, as it has been postulated, there should be a population of ‘editors’ or competent code users. In particular, it was noticed that editing any code or language is not performed via the selection of errors, as “no natural language speaks itself as no natural codes itself”. They result out of social interactions of agents competent to edit a language or a code in a context-dependent way [39]. Thanks to their ability to generate new sequences (produce diversity) and the ‘social’ (co-operative and counteractive) behavior of quasispecies [40], it was argued that viruses constitute a population of such editors. In terms of the DNA code, point mutations, followed by selection of the fittest type, are not sufficient for the formation of a new network as this process requires editing preexisting code and assigning it a new meaning in a new context. The mixed population of viruses (quasispecies) was proposed to have all the competences to edit and extend the existing code [21,27].

## 4. ‘Big Bang’ of an Adaptive Immune Response in Jawed Fish

The adaptive immunity network, with all its components—recombination activating genes RAG1 and 2, immunoglobulins, T cells with their receptors (TCRs) and MHC class I and class II molecules—first appeared in jawed fish [41,42,43,44]. Despite similar surface recognition systems being identified in jawless vertebrates, they are not related to the components of the adaptive immunity. Since there is no living organism resembling the intermediate of this process [23,26], the outbreak of adaptive immunity in jawed fish has been described as a “big bang” event. Interestingly, invertebrates like *C. elegans* or shrimps do have effective antiviral systems based on RNA interference (RNAi) that appear to effectively fight any type of viral infection [45,46,47,48,49]. On the contrary, vertebrate immune cells, despite being effective in fighting viruses, are themselves prone to viral infections. Their white blood cells, specific to adaptive immunity, do support a plethora of viruses not known to infect many ancestral species including: negative-stranded RNA viruses (vesicular stomatitis virus, parainfluenza virus, measles virus, respiratory syncytial virus); retroviruses (murine leukemia virus [MLV], endogeneous type C viruses, FeLV, HIV, HTLV); single-stranded DNA viruses (porcine parvovirus, minute virus of mice); and double stranded DNA viruses (papovaviruses, group C adenoviruses, eight distinct lineages of human herpesviruses, and leporipoxviruses) [26,50]. Such observation is contradictory to the view that the highly sophisticated system of adaptive immunity is more capable of fighting viral pathogens. If we, however, according to Villarreal, think of the adaptive immunity as a system of cellular identity based on addiction modules, the colonizing populations of viruses could explain the origin of its components (genes and regulation) [26]. The immune cells of jawed vertebrates, unlike their ancestors, are educated on self-identity during immune system development (immune education), which allows them to recognize virus-infected self-cells, proliferate, and destroy them.

Comparative genomics between jawless and jawed fish provided clues regarding the genetic alternations associated with the origin of adaptive immunity: the jawed fish underwent a large-scale viral colonization of the genome by several families of ERVs, and a huge genome expansion.

Interestingly, in compact genomes of invertebrates, the conserved low number ERVs and LTRs are also present, but they are different than vertebrate-like ERVs and LINES (long interspersed sequences, retroviruses-related transposable elements) and are largely outnumbered by non-LTR DNA transposable elements [26]. There are also large differences regarding the amount and the makeup of parasitic elements between different invertebrate groups, which appear to be associated with the virosphere composition. Compared to non-marine invertebrate *C. elegans*, the shrimps’ DNA is rich in species-specific satellite sequences (up to 100,000/genome), which, despite residing in the heterochromatin region, can be transcribed during the shrimps’ development [51,52]. Some species of shrimps also have an unusually high number of pseudogenes in their mitochondrial genomes, suggesting retroposon action during evolution [53]. Marine invertebrates like shrimps or mollusks, despite the lack of immune cells, do support many types of viruses (though retroviruses are rare). Thus, it has been suggested that shrimp (and other invertebrates) genomes have been largely shaped by the action of species-specific genetic parasites, but the makeup of these parasites is distinct to those seen in jawed fish. The majority of DNA transpozons appear to have been lost with the emergence of vertebrates.

Members of Chordata phylum, tunicates, are the closest living invertebrate relatives of vertebrates. Their very compact genomes, with around 15,000 genes, can fuse with one another while mixed individuals are rejected. Rejection occurs via elimination of one partner in a chimera. It is an immune cell-based reaction and the principal cell type to mediate partner elimination is a cytotoxic morula cell (MC), which in collaboration with activated phagocytes eradicate chimeric partners [54]. It more resembles the immune response of natural killer (NK) cells than cytotoxic T lymphocytes and is not antigen specific. The Fu/HC locus of tunicates also does not appear to protect against pathogens, but rather acts against other tunicate colonies. Interestingly, the RAG genes, crucial for the generation of the genetic diversity of immune receptors have not been found in the tunicate representative sea squirt, but have been identified to function as a DNA transposase in the genome of echinoderms (invertebrates), though not as part of an immunity (identity) system [55,56]. It has been thus concluded that RAG-related elements have independently colonized invertebrate and vertebrate genomes, but not their respective ancestors. This supports the concept that the adaptive immunity components were acquired via a complex genetic colonization event by a mixed population of viruses [26].

Jawless lampreys and hagfish constitute the linage that diverged before jawed fish and are grouped in cyclostome taxon [57,58]. As the immediate ancestors of jawed vertebrate, jawless fish have been studied extensively in search for the origin of the adaptive immune system [59,60,61,62,63]. However, none of the components of jawed vertebrate adaptive immunity (immunoglobulins, T cell receptors, recombination activating genes RAG1 and 2, and MHC class I and II molecules) have been identified in this taxon. Interestingly, jawless fish have been shown to possess lymphocyte-like cells [64] and an equivalent of vertebrate immunoglobulin and antigen receptors, namely variable lymphocyte receptors (VLRs) [65,66]. However, strategies for creating diversity in the lymphocyte receptors of jawed and jawless vertebrates are different. While jawed vertebrates rearrange variable, diverse, and joining gene segments (VDJ) to produce diverse repertoires of B cells and T cell antigen receptors, hagfish and lampreys use another approach. Hagfish have one incomplete copy of the *VLR* gene and lampreys have two such incomplete genes (*VLA-A* and *VLA-B*) [43,67]. They are flanked with multiple leucine-rich repeat (LRR) modules. A diverse repertoire of cell surface VLRs is created through somatic rearrangement and insertion of LRR cassettes into an incomplete germline *VLR* gene [65]. Though both jawless and jawed vertebrates produce antigen receptors with anticipatory activity toward antigens, they have different evolutionary origin. The VLRs of jawless fish, thus, cannot be considered as intermediates in the evolution of jawed vertebrate antigen receptors. Moreover, in a contradiction to vertebrate lymphocytes, the lymphocyte-like cells of lampreys and hagfish do not support any type of retrovirus [26]. Their genomes are also much less colonized with retroviral elements and their derivatives (long and short interspersed elements: LINES and SINES), and lack all of the core elements of adaptive immunity like T cells with their receptors (TCRs), the RAG1/2 immunoglobulin rearrangement or major histocompatibility complex (MHC) genes, allowing for non-self-recognition, clonal proliferation, and killing of non-self-cells.

A cartilaginous fish representative, the nurse shark, is the earliest vertebrate with a complete repertoire of adaptive immune system components [68,69]. The appearance of this system is clearly associated with large-scale ERV colonization (e.g., gypsy-like chromoviruses), expansion of reverse transcriptase (RT)-dependent elements (LINEs, SINEs), large scale duplication of the genome, and an elimination or inactivation of DNA transpozons. It has been argued that this extensive colonization of fish genome resulted in major changes in the genetic identity of the host and established transformed blood cells with anti-viral activity (to oppose related viruses according to an addiction module mechanism). The products of ERV colonization allowed them to recognize and kill virus-infected self-cells. Historically, a huge increase in chromoviral elements (both ERVs and their defectives, LTRs) in bony fish has been dismissed and explained by the product of overactive selfish DNA with no functional consequence. This explanation, however, does not clarify why these elements expanded and, importantly were maintained in fish, plants, and insect genomes, but were subsequently removed from mammalian genomes. However, if we consider the perspectives that these parasitic elements provided genes and regulation crucial for establishing the adaptive immunity network (via addiction module mechanism) and that subsequent viral colonizations enhanced this network (also via displacing or inactivation of elements from previous colonization events), the positive selection for virus-derived elements can be explained [26]. In particular, the fact that the virus that stably colonizes one population preserves its lytic potential (a toxic element of an ‘addiction module’) and can be transmitted to another uncolonized population (and kill it), provides the rationale of why the colonized population wins. An important fact to consider here is the basic version of the adaptive immunity genes (e.g., the TCR receptors) can be found in viruses) [26]. The widely accepted view is the virus piracy concept: viruses simply steal the host genes and transfer them between different hosts like vehicles, so the viral genes originally come from hosts. However, this hypothesis has been challenged with the observations that: (1) many of the viral genes are unique to viruses and (2) when viral genes show similarity to the host genes, it is usually the viral gene that is ancestral to the host [70]. Phylogenetic analysis indicates that the most basal members of TCR-related genes are the viral *JAM/CTX/PVR* genes, also called the CTX-like family of viral receptors [71]. They support many virus interactions and are often used as specific virus receptors (PVR-polio virus receptor; JAM-reovirus receptor). This suggests the viral source of T-lymphocyte receptors [26].

## 5. Major Histocompatibility Complex (MHC) Identity System

### Major Histocompatibility Complex (MHC) Locus and Link to Odor Receptors

The major histocompatibility complex (MHC) locus is an approximately 3.6 Mb segment located on the short arm of chromosome 6 (6p21). It is one of the most “dense” regions in the human genome as it consists of more than 300 loci with over 160 protein-coding genes involved in the innate and adaptive immune responses, transcription regulation, and signaling factors [72,73]. The MHC region is divided into three sub-regions: MHC class I, class II, and class III. Together with other components of adaptive immune response, MHC I and II first appeared in cartilaginous fish and are found in all jawed vertebrates [42,74]. Human MHC class I contains HLA I genes (classical: *HLA-A*, *HLA-B*, *HLA-C* and nonclassical: *HLA-E*, *HLA-F* and *HLA-G* molecules) and genes involved in antigen presentation. MHC class II contains HLA II genes (*HLA-DPA1, HLA-DPB1, HLA-DQA1, HLA-DQB1, HLA-DRA*, and *HLA-DRB1*) (Figure 2) [75]. They encode for molecules presenting antigenic peptides to T cells, thus constitute a part of an identity system essential in T cells education (to set self-identity and protect from self-destruction) and allowing for non-self-elimination. MHC class III (HLA class III in humans) is a name given to a cluster of genes between class I and class II, which encodes proteins involved in various processes including heat shock proteins (HSP70), TNF family proteins (TNF-α, TNF-ß), and complement components C3, C4, and C5 [76,77]. Approximately 50% of HLA gene sequences consist of interspersed repeats including SINEs (Alu, MIR), LINEs (LINE1 and 2, L3/CR1), LTR elements (ERV-L, ERV class I, and class II), and DNA elements [75]. Furthermore, more than 1500 microsatellites have been identified in this region.

HLA gene cluster is characterized by gene density, high polymorphism, and strong linkage disequilibrium (LD) [78]. According to the IPD-IMGT/HLA database, unique 19,031 HLA class I alleles and over 26,000 allele sequence for other HLA genes within the MHC region has been identified (assigned as of January 2020) [79]. The highest polymorphism is found in exons 2 and 3 that encode two extracellular domains (α1 and α2 domains, respectively), which together form a peptide-binding groove. The MHC variability in the peptide binding site allows for the binding of a wide range of antigens, thus providing the immune system with an advantage against pathogen diversity. Moreover, the codominant expression of MHC alleles contribute to a greater number of MHC class I molecules being expressed in the cell. Polygeny of the MHC locus, together with an extreme polymorphism and a codominance, result in a great diversity of MHC molecules present on each cell and within the population.

The genome-wide association studies (GWAS) have linked a variation within HLA genes with the higher incidence of infectious and autoimmune diseases such as rheumatoid arthritis, psoriasis, and asthma [80,81]. Furthermore, recent studies indicate that the loss of heterozygosity of the HLA locus was associated with worse prognosis in patients with non-small lung cancer and glioblastoma [82,83] and worse response to cancer immunotherapies [84]. Farh and colleagues showed that ~90% of causal autoimmune disease variants reside within non-protein-coding regions of the human genome, with ~60% mapping to immune cell enhancer-like elements, which gain histone acetylation following immune stimulation, hence contributing to the transcription of non-coding regulatory RNAs [85]. These regulatory RNAs usually arise from viral derived elements (LTRs, LINES, Alus), which are densely distributed in the MHC region [86,87,88,89].

In non-primate tetrapods, the MHC locus is linked to odor receptor (OR) genes, involved in offspring and mate identity. One class of OR genes, vomeronasal receptor (VNO), binds pheromones affecting family and sexual behavior. In humans, all five open reading frames (ORFs) of the *VNO* gene family have been converted to pseudogenes [90,91,92,93]. Loss of the linkage between MHC and pheromone receptors in primates suggests a significant change in the determination of self-identity (decreased olfaction dependence), which seems to have resulted from a large-scale ERV colonization [23,26].

## 6. Viral Elements-Mediated Evolution and Regulation of MHCI Expression

Multiple independent human ERV (HERV) colonizations have resulted in 30–50 HERV families inhabiting the human genome, the majority in non-protein coding regions [94,95,96]. Since the original function of MHC as a system of identity was to identify and fight virus-infected self-cells by generating active T cells, the fact that the MHC region has been particularly shaped by these colonization events can be viewed as a consequence of a viral-mediated enhancement of self-identity (and immunity) [26]. As a result, the MHC region has become increasingly complex during evolution from vertebrates to mammals [97,98,99,100]. The density of HERVs and retroelements within the MHC region is 10-fold greater than in other regions of the chromosomes with as many as 16 HERV elements localized within the MHCI locus [101,102]. During mammalian and primate evolution, the MHC region went through various genomic rearrangements including crossover events of *MIC* (MHC class I chain-related), *HERV16*, and *HLA* class I genes, which influenced the structural organization of MHC [103]. It seems that *HERV16* (repeated at least twelve times), along with *HLA* class I coding and noncoding sequences, has been a recombination site for many of the duplication events, like unequal crossovers [102,103].

The MHC class I molecules present a tissue-specific expression pattern, being expressed at the basal level in nearly all nucleated cells, however, the HLA abundances vary markedly between various tissues. Recently, Bögel et al. reported up to a 100-fold difference in intra-tissue median HLA abundances. The highest levels of classical HLA class I, class II, and non-classical HLA class I molecules were found in cells of the immune system and lymphatic organs, whereas immune-privileged tissues such as brain, retina, and muscles showed the lowest expression [104,105]. Based on the immunomodulatory properties of HLA, its tissue-specific and inducible expression should be under tight transcriptional regulation. For a long time, the quest to unravel MHC regulation was oriented towards finding a ‘master transcriptional regulator’ that would govern the regulatory complexity behind MHC expression. Although several regulatory proteins have been identified, they do not seem to be sufficient for the complex spatio-temporal regulation of MHC class I expression. It is more probable that the complex MHCI regulation, which needs to be coherent with a broader adaptive immunity network, can only be attained thanks to regulatory non-coding RNAs. Below, we briefly describe the classic view on the protein-based MHC class I regulation, followed by the emerging impact of the network of non-coding RNAs.

The thorough analysis of the MHC class I gene promoter has identified distinct highly conserved cis-acting regulatory DNA elements located in the proximal and distal regions upstream of classical and non-classical MHCI genes (Figure 3). Many regulatory proteins that bind to these elements have also been identified. However, even together with the recent finding of what initially was thought to be a master class I transactivator—the NLRC5 protein [106]—they do not solve all *MHCI* regulation puzzles. Though NLRC5 was indeed shown to regulate basal expression of MHC class I molecules and antigen processing and presentation genes (*TAP1, LMP2*) in immune cells, its contribution to MHCI expression in other cell types has been rather modest. Additionally, despite the basal level of MHCI being reduced in *NLRC5*^−/−^ T and B cells, IFN gamma induced *MHCI* in these cells, indicating the presence of compensatory mechanisms regulating *MHC* class I expression [107,108,109]. Thus, MHC class I does not seem to be under the control of a master transcriptional regulator, as observed for MHC class II, where CIITA (class II transactivator) and RFX proteins are crucial for the expression of class II molecules. Mutations in *CIITA* or any of the four RFX regulatory genes (*RFXANK, RFX5*, and *RFXAP)* cause bare lymphocyte syndrome type II (BLSII), an immunodeficiency characterized by the lack of MHC class II expression. Knockout mice lacking *CIITA* showed reduced or absent class II expression on dendritic cells and B lymphocytes and the inability of IFN gamma (IFNg) to induce class II [110,111].

MHC class I gene expression is mediated (in an inducible manner) by NF-κB binding to enhancer A and IFN-stimulated response element (ISRE) binding motifs located within the promoter regions (Figure 3). In particular, the classical MHC class I genes *HLA-A* and *HLA-B* can be stimulated by NF-κB and are highly induced in response to IFN gamma [117,118]. The in vitro studies showed that NF-κB-induced MHC class I expression plays the most important role for the *HLA-A* locus, which contains two NF-κB binding sites in its Enhancer A region. In contrast, *HLA-B* contains only one NF-κB binding region supported by a Sp1 binding site (Sp1 has been shown to interact with NF-κB) [115,119]. IFNg-induced expression is mediated by the binding of interferon response factors (IRFs) to ISRE [120]. IFN gamma induces the expression of IRF-1 via the JAK/STAT pathway by the activation of Janus kinases (JAK) 1 and 2 and phosphorylation of STAT1. The DNA sequence of ISRE varies among HLA class I loci, resulting in locus-specific differences in the IFN-induced MHC class I [121,122]. *HLA-A* gene has a weaker response to IFN-gamma induction than *HLA-B* and *HLA-C*, which has been explained by differences in the ISRE structure [123,124].

Recently, an important finding regarding the regulation of the IFN gamma network has been reported [29]. Chuong and colleagues have found that LTR promoter regions of ERV family MER41, an endogenized gammaretrovirus that invaded the genome of an anthropoid primate ancestor ~45–60 million years (MY) ago, constitute binding sites for IFN gamma-regulated transcription factors STAT1 and IRF1. The chromatin immunoprecipitation followed by sequencing (chip-seq) analysis in HeLa cells revealed that, upon IFNg stimulation, the MER41B sequences had been enriched within STAT1 peaks and contained a tandem pair of Gamma Activation Site (GAS) motifs predicted to bind STAT1 in response to IFNg. Deletion of the MER41 element from the promoter region of the *Absent in Melanoma 2* (*AIM2*) gene (a sensor of foreign DNA, stimulated by IFNg and activating an inflammatory response) in HeLa cells abolished an IFN gamma-induced reporter expression in the luciferase reporter assay. Moreover, the deletion of MER41 also abrogated inflammasome activation after vaccinia virus infection. Further analysis showed that different mammalian lineages were independently colonized by related MER41-like gammaretroviruses ~50–75 MY ago. Remarkably, the authors found that STAT1 binding motifs, present in the primate specific MER41 family, are conserved in MER41-like relatives in lemuriformes, vesper bats, carnivores, and artiodactyls. Reconstructed ancestral consensus sequences from MER41-like LTRs from cow and dog could indeed drive IFNg-inducible reporter activity in HeLa cells. This study showed that families of related ERVs have independently expanded the IFN regulatory network in multiple mammalian lineages as a result of numerous independent retroviral colonization events. The essential role of species-specific persistent viruses in building an immune response network has been therefore confirmed, and together with the fact that virus-derived elements are responsible for placenta formation [14] and building the p53 network (as binding sites for p53 transcription factor) [30], it becomes evident that the action of viral-derived elements has been essential in the establishment and enhancement of complicated networks. 

So far, the studies on ERVs in the context of the regulation of the immune system have revealed their role as enhancers—regulatory sequences for binding immune transcription factor, though it has lately become apparent that the majority of human ERVs are transcribed to non-protein coding RNAs [88]. The crucial characteristic of ncRNAs is their extraordinarily specific expression, both in terms of space (specific cell and subcellular organization) and time [125]. ncRNAs have been shown to be particularly active in controlling developmental processes and orchestrating complicated networks like the human brain or immune response [126,127,128,129]. Single nucleotide variants (SNV) in ncRNAs have also been associated with multiple diseases. There is still much to learn about viral element (ERVs, LINEs, SINEs, Alus)-derived ncRNAs, their interactions and function, but according to Villarreal and Witzany, these gigantic networks of interacting RNAs are expected to have derived from addiction module action (i.e., results from persistent colonization events and brings a more complex RNA-based system of identity and immunity, mainly acting via the miRNA network) [19,22,31]. Alu elements are derivatives of ERVs, estimated to have inserted into 75% of all human genes [130]. They were shown to provide miRNA target sites to the 3′ untranslated regions (UTRs) of numerous transcripts [86,87]. The abundance of Alu sites and their ability to distribute miRNA targets were suggested to allow for the establishment and extension of complex regulatory networks. Interestingly, the expression of Alu transcripts is regulated by a virus infection. Normally low levels of polymerase III transcribed ‘sense’ Alu transcripts increased dramatically upon infection with Adenovirus type 5 (Ad5) and herpes simplex virus type 1 (HSV-1). The increased expression of sense and antisense Alu transcripts should result in dsRNAs, causing translation inhibition and providing an antiviral response, like the interferon pathway [19]. It suggests that an induction of Alus transcription may be a part of a normal cellular response to a viral infection. A similar observation was made with LINE-2 (L2) elements, which entered the human genome around 100–300 MY ago, and have also been proven to give rise to miRNAs [131,132]. L2-miRNAs are derived from the 3′ end of the L2 consensus sequence and thus share very similar sequences, indicating that L2-miRNAs could target transcripts with L2s in their 3’UTR. In line with this, it has been observed that many protein-coding genes carry fragments of L2-derived sequences in their 3’UTR and these sequences serve as L2-miRNAs target sites. Moreover, L2-miRNAs and their targets are generally ubiquitously expressed at low levels in multiple human tissues, indicating a role for this network in buffering transcriptional levels of housekeeping genes. This network was shown to be perturbed in glioblastoma [132]. There is thus no doubt that virus-derived parasitic elements have established and extended a post-transcriptional network that shapes transcriptional regulation. We should also acknowledge that this process is still ongoing [20].

In line with this observation, parasitic elements of the MHC class I region give rise to a great number of miRNAs. In silico analysis revealed thousands of putative pre-miRNA loci that could be expressed from the MHC region by different cell types and at different developmental stages, though only a few have so far been analyzed experimentally [133]. miR-6891-5p has been shown to originate from a highly conserved intronic segment of the *HLA-B* gene and regulate the expression of numerous immunologically related transcripts including those encoding the heavy chain of IgA [134,135]. Deep sequencing of two lymphoblastoid cell lines with a fully characterized MHC haplotype identified 89 new miRNAs, 43 of which lie within linkage disequilibrium blocks that contain disease-associated single nucleotide polymorphisms (SNPs). These SNPs correlate with 65 unique disease phenotypes, suggesting the putative role of these miRNAs in the etiology of numerous diseases associated with the MHC system [133]. 

Apart from miRNAs interfering with gene expression, there are also other regulatory RNAs shaping MHC (and other) genes expression in different ways. A widely studied long ncRNA (lncRNA) encoded within the MHC class I locus is a human-specific *HCP5* RNA gene (*Human Leukocyte Antigen pseudogene P5*) [136]. It is located between the *MICA* and *MICB* genes, centromeric of *HLA-B*. Interestingly, *HCP5* encodes an antisense transcript containing 3’LTR and *pol* sequence of *HERV16* linked to a *HLA* class I promoter and leader sequence [89,102]. It has evolved from an ancient ERV insertion (HERV16, ~37 MY ago) and expanded the regulatory function by sequestering the MHC promoter and enhancer region from an ancient *HLA* class I gene. This unusual hybrid structure of the transcript accounts for its multiple functions. The *HERV16* insertion in the *HCP5* transcript is linked to the promoter and gene fragment of *HLA* class I (exon 1), which regulates its expression. Therefore, *HCP5* transcription should be concomitant with *HLA* class I genes [89]. Indeed, their expression is often coordinated, but in some cases can be reversely regulated (*HCP5* is upregulated while *HLA* class I genes are downregulated) due to epigenomic regulation or the *HCP5* interactions with miRNAs [137]. Due to the antisense orientation and complementation with viral LTR and *po*l sequences, *HCP5* was initially believed to hybridize and destroy viral RNA [138]. Yoon et al. experimentally showed that the mechanism of *HCP5* action was not that simple [139]. Accumulating evidence indicates that *HCP5* interacts with various miRNAs to either reduce or enhance their regulatory effect on target mRNAs. The competition or cooperation between this lncRNA and miRNAs in regulating gene expression seems to be context dependent. Since the origin of the *HCP5* lncRNA is an ancient retroviral insertion and its action affects cell survival and immunity, it can be viewed as an example of a virus-mediated enhancement of immunity (identity) resulting from an addiction module action.

## 7. MHC-Regulating Non-Coding RNAs in Cancer

In cancer, *HCP5* was found to compete with tumor suppressive miRNAs and drive the expression of oncogenes [89]. In the basal subtype of breast cancer, *HCP5* was bound to miRNA-155, affecting cell proliferation and aggressiveness of tumor [140], whereas in glioma, malignancy was shown to be regulated by an *HCP5*-miRNA-139-*RUNX1* feedback loop. miRNA-139 acts in a tumor suppressive fashion, downregulating Runt-related transcription factor 1 (*RUNX1*) gene expression. Oncogenic RUNX1 promoted the growth of glioma cells as well as the expression of *HCP5* and its binding to the tumor suppressor miRNA-139. The “sponging” effect of *HCP5* on miRNA-139 unleashed the *RUNX1* expression, connecting *HCP5* to glioma oncogenesis. *HCP5* was also found upregulated in glioma tissues as well as in U87 and U251 glioblastoma cell lines [141]. 

There are more studies showing the involvement of *HCP5* lncRNA in innate and adaptive immune response, and associating its expression with some autoimmune diseases and cancer [89]. Yuan et al. have shown that 6p21 and 15q25 loci, which are lung cancer risk-related loci, were enriched in lncRNAs including *HCP5*, *RP11-650L12.2*, *XXbac-BPG27H4.8*, and *HCG17*. Analysis of 17,153 cases and 239,337 controls pointed out that at least six *HCP5* SNVs (including rs3130907 within the *HCP5* sequence) were significantly associated with lung cancer susceptibility [142]. Previously, Orvis et al. showed the influence of the inactivation of the *BRG1* gene that encodes the ATPase subunits of the SW1/SNF chromatin remodeling complex, on downregulation of the *HCP5* expression, and all of the classical and nonclassical HLA class I genes. This inactivation of the *BRG1* gene also contributed to the aggressiveness of non-small cell lung cancer [143]. Additionally, *HCP5* was positively correlated with a poor prognosis of lung adenocarcinoma patients, especially ones with *EGFR* and *KRAS* mutations [144]. Moreover, transcription factor SP1 induced upregulation of *HCP5*, promoting the development of osteosarcoma [145]. Lee et al. showed first evidence describing the function of special AT-rich sequence binding protein 1 (SATB1) in lncRNA transcription including *HCP5*. SATB1 was found to be highly expressed in aggressive breast cancer, and it also enhanced *HCP5* in these cells [146]. Significantly, *HCP5* was found to be overexpressed in lymph node metastasis of different types of cancer including small cell lung cancer [147,148], colorectal cancerous tissue [149], glioma tissue [141], and prostate cancer [150]. Based on these observations, it has become apparent that *HCP5* could be targeted for knockdown in antitumor therapeutics.

As previously mentioned, virus-encoded miRNAs allow for the stable installation of the virus in the host cell. They counter-regulate the lytic virus potential (often by inhibiting apoptosis, thus making the cell addicted to the virus presence) and trick the host immune system in the way that the virus-infected cell stays unnoticed. The network of regulatory RNAs is cell type and developmental-stage specific. Since in tumors the expression and interactions of regulatory RNAs are out of balance, numerous miRNAs have been associated with different cancer types. Below, we can observe a few examples of associations between miRNAs regulating *MHC* and *APM* genes and their correlation with different tumor types.

Studies in nasopharyngeal carcinoma cells connected the miRNAs with the expression of antigen-processing machinery (APM) components. Gao et al. demonstrated that miRNA-9 controls the expression of the classical MHC class I pathway through targeting proteasome subunits *PSMB8* and *PSMB10*, *TAP1*, *β2M*, *HLA-B*, *HLA-C*, and the nonclassical *HLA-F* and *HLA-G* antigens [151]. The binding of miRNA-9 to the 3′-UTR of these molecules has not yet been shown, however, despite this, the regulation of APM deficiencies mediated by miRNA-9 might be responsible for tumor immune escape [152]. Furthermore, miRNA-148a has been shown to bind to the *HLA-C* 3′-UTR, and consequently affect *HLA-C* expression on the cell surface and T/NK cell responses [153,154]. Mari et al. proved that miR125a-5p and miR148a-3p can regulate *TAP2* and *MHC* class I expression in esophageal adenocarcinoma cells, thus suppressing an anti-tumor immune response, leading to poor outcome of patients [155].

Since the aberrant expression and/or interactions of regulatory RNAs in cancer impact MHC gene expression, they do have an influence on different HLA class I altered phenotypes (Table 1) and consequently should be included in the disrupted expression of HLA class I molecules in cancer. Therefore, it is clear that noncoding RNA interactions modulate physiological functioning of the immune system and their dysregulation might contribute to cancer occurrence.

## 8. MHC Class I Alterations in Tumors

Loss or reduced expression of MHC class I molecules on cell surface is one of the most important immune-escape mechanisms in tumors, leading to resistance to T cell cytotoxicity. MHC class I molecules are crucial in antigen presentation to T cells and modulation of NK activity. Therefore, alterations in MHC class I expression in tumors have a high impact on the immune response. These alterations can occur at many different levels of MHC class I expression including transcriptional, post-transcriptional, genetic, and epigenetic levels [171]. There is a wide spectrum of the percentage of MHC class I loss in tumors, mostly depending on the tumor types. The highest loss of HLA class I was observed in cervical [157] and in breast carcinomas [172], where the rate was 96%, but also in colorectal carcinoma, with the rate of 87% [173], and in laryngeal carcinomas, with the rate of 70% [174]. The defects in HLA class I expression are highly connected to tumor progression and poor patient prognosis [175].

The expression of HLA class I alleles in tumor cells highly differ among patients and in various tumor types, making the evaluation of HLA class I molecules very complex [176]. Recently, Kennedy et al. broadened the existing knowledge on the complexity of HLA class I expression and their different organization in the cell surface membrane. They showed that HLA-C, when compared to HLA-B, forms larger and more numerous clusters at the cell surface. Moreover, HLA class I molecules are shown to be differently organized on the surface of primary cells. B cells have a more homogenous organization compared to a more clustered organization in the case of NK and T cells [177].

### 8.1. HLA Class I Altered Phenotypes

Based on the different mechanisms and molecular defects appearing in MHC class I expression in tumors derived from different tissues, there are seven major HLA class I altered phenotypes (Table 1): (i) phenotype I—HLA class I total loss; (ii) phenotype II—HLA haplotype loss; (iii) phenotype III—HLA-A, -B or -C locus product downregulation; (iv) phenotype IV—HLA allelic loss; (v) phenotype V-compound phenotype; (vi) phenotype VI-unresponsiveness to interferon; and (vii) phenotype VII–absence of classical HLA molecules (Ia) with aberrant expression of non-classical HLA molecules (Ib) [178].

### 8.2. Role of HLA Class I Alterations in Immunotherapy

The lack of tumor rejection is associated with multiple cancer immune escape mechanisms including the loss or low expression of tumor HLA class I molecules. The absence of the normal expression of HLA class I molecules on tumor cell surface leads to tumor progression [179]. The impact of MHC-I defects on the non-responding tumors is largely unknown and corrections of antigen presentation in these tumor types might result in a much higher success rate of immunotherapies [180].

Based on the accumulating evidence, tumor MHC class I expression influences the degree and composition of immune cellular infiltration, and consequently T cell response during cancer progression. Therefore, the success of traditional and newly developed immunotherapy approaches greatly depends on MHC class I expression on tumor cells. Recently, Chowell et al. showed that both patient-specific HLA-I genotype as well as somatic alterations in tumors affected the clinical outcome of immune checkpoint blockade, suggesting that these factors could be considered in the design of future clinical trials [84]. Additionally, their findings indicate that alternative ways to harness the immune system may be necessary since HLA-I homozygosity and LOH at HLA-I were associated with decreased overall survival in patients treated with immune checkpoint blockade, and by that, represented a genetic barrier to effective immunotherapy [84]. Perea et al. had similar observations while analyzing the density and composition of tumor T-cell infiltration in non-small-cell lung carcinoma in relation to PD-L1 and HLA class I expression. They observed that positive HLA-I expression, independently of PD-L1 status, is the key factor determining the increased density of immune infiltrate. Accumulating evidence shows that although PD-L1 expression has been associated with cancer immune response in some patients submitted to immunotherapy, tumor HLA-I expression is the crucial factor driving the positive response and determining the efficacy of the therapy [181].

According to Aptsiauri et al., tumors with irreversible alterations in HLA class I can escape the immune system despite immunotherapy, leading to the assumption that tumors with reversible alterations will respond better to immunotherapy by upregulation of the antigen presenting machinery, consequently leading to tumor recognition and elimination by T cells [179]. Similarly, Carretero et al. observed a strong correlation between tumor progression/recurrence and response to therapy with defects in tumor HLA class I expression in melanoma [182] and bladder cancer [183].

Therefore, based on accumulating evidence, we can conclude that the expression of HLA class I alterations in tumor cells should be considered during the selection of immunotherapy strategies as well as monitored during treatment as a potential biomarker [179]. Moreover, recovering MHC class I expression on tumor cells, together with a proper immunotherapy approach, might lead to better patient prognosis and survival [176].

## 9. Conclusions

This review brings about the idea stated previously by Luis Villarreal that, counterintuitively, immunity (identity) is likely to have been invented by viruses. According to this concept, the persistent virus–host relationship is a driving force of both virus and the host evolution and leads to the formation of the new virus–host entity (with new genetic identity and an enhanced immunity), which is better adapted to living in its viral habitat. Persistence is acquired via stable installation of the virus (or virus-derived element) in the host cell by a mechanism described as an ‘addiction module’, where the killing (lytic) effect of the virus is counter regulated by an anti-killing feature of the same virus (e.g., inhibition of apoptosis) and complemented by immune evasive viral mechanisms. In Eukaryotic cells, where the expression of genetic material is regulated by RNA-mediated processes, viral persistence is most often acquired via virus-encoded regulatory RNAs. Indeed, persistent DNA viruses, ancient endogenized viruses (ERVs) and their derivatives (LTRs, LINEs, SINEs) are the source of regulatory non-coding RNAs such as lncRNAs or miRNAs. As the parasitic virus-derived elements interact with each other and easily disperse over the genome, they establish and extend gene regulatory networks. Regulatory RNAs encoded by these viral-derived elements usually control cell proliferation and immunity as they are used in the establishment of the persistence via the addiction module mechanism. These networks of regulatory RNA interactions are imbalanced in cancer, which is why many non-coding RNAs associate with tumors and the cancer- associated SNPs reside within regulatory RNA transcripts. MHC class I downregulation is a feature of many cancer types, allowing for tumor progression and resistance to immunotherapies. In this review, we show that the complex regulatory network behind MHC expression is largely controlled by virus-derived elements, both as binding sites for immune transcription factors and as sources of regulatory RNAs, like miRNAs. The capacity to use these regulatory RNAs as diagnostic or therapeutic targets in tumors seems very promising. In particular, an increased expression of *HCP5* – HERV16-derived long non-coding RNA of oncogenic potential in various cancer types could be considered as a cancer diagnostic marker. Since *HCP5* drives the expression of different oncogenes, it could also be targeted for knockdown in anticancer therapeutics.

## Figures and Tables

**Figure 1 cancers-12-01155-f001:**
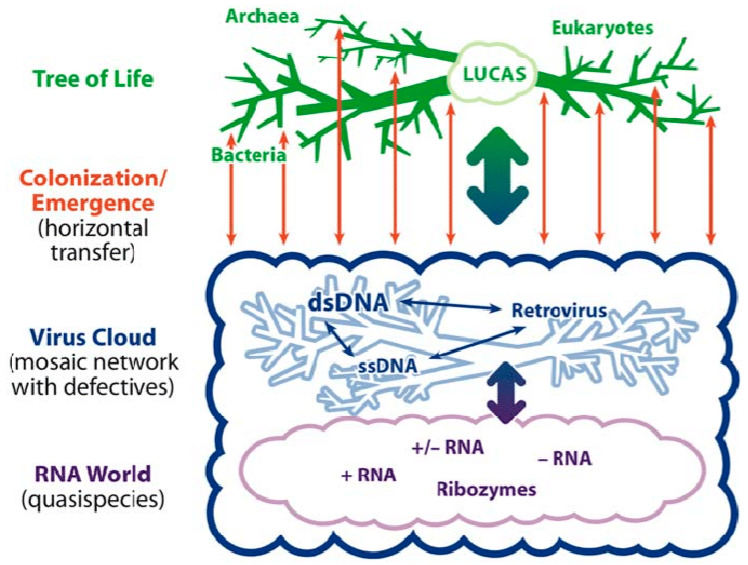
Representation of the relationship of the Tree of Life (green dendrogram) to the virosphere (blue cloud). The blue dendrogram represents species-specific persisting viruses. Reprinted from [21] with permission.

**Figure 2 cancers-12-01155-f002:**
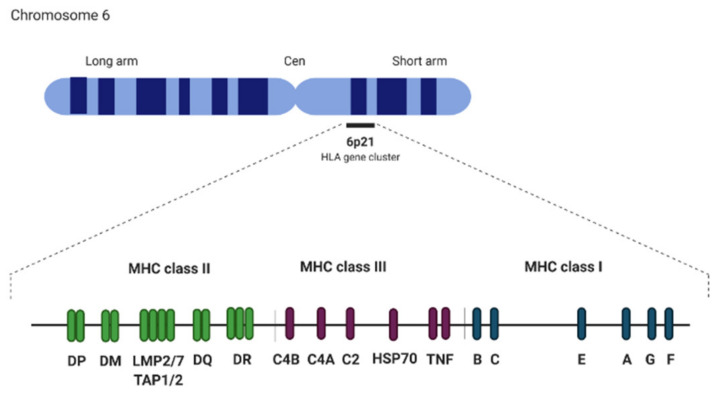
Major histocompatibility complex (MHC) locus structure.

**Figure 3 cancers-12-01155-f003:**
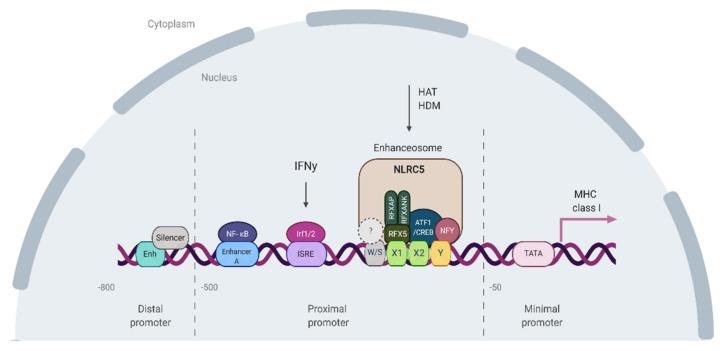
The MHC class I promoter region. Tissue-specific expression of MHC class I is mediated through the regulatory elements located in the distal promoter (overlapping enhancer and silencer located −800 and −700 bp upstream to the transcription start site) [112]. The basal transcription level is mediated by the minimal MHC class I core promoter sequence (located from −50 bp to +14 bp) [113]. The minimal core promoter is comprised of elements required for RNA polymerase complex binding including CCAAT box, a TATA-box, an Initiator (Inr)-like motif, and a Sp1-binding site. The inducible and constitutive expression of MHC is regulated by the proximal promoter elements, located −500 bp–50 bp upstream to the transcription start site. At the DNA level, the proximal promoter is comprised of four sequences (W/S, X1, X2, and Y boxes) called the ‘SXY module’, which is found in all MHC class I genes and is highly conserved across vertebrates [114,115]. The X1 box is bound by regulatory factor complex X (RFX), a constitutively expressed trimer composed of RFX5 (Regulatory Factor X 5), RFXAP (RFX-Associated Protein), and RFXANK (RFX containing three Ankyrin repeats). The X2 box and Y boxes of the SXY module are bound by cAMP-responsive element-binding protein (CREB), activating transcription factor 1 (ATF1) [116], and nuclear transcription factor Y (NFY) respectively. The factor(s) that interact with the W/S box are still to be determined. This complex set of protein-DNA and protein–protein interactions is crucial for the recruitment of the transactivator protein NLRC5 and the formation of an enhanceosome. HAT–histone acetylase; HDM–histone demethylase. Created with BioRender.

**Table 1 cancers-12-01155-t001:** HLA class I altered phenotypes.

Phenotype	Characteristics	Description
I	Total loss of HLA class I molecules	Low frequency in laryngeal carcinomas (10%), colorectal carcinomas (18%), and melanomas (17%), and higher in breast (52%), prostate (40%), and bladder (35%) carcinomas [156].
II	Loss of an HLA class I haplotype	Produced by loss of heterozygosity (LOH) associated with chromosome 6.Incidence of this altered phenotype is 46% in cervix carcinomas, 15–49% in head and neck, 17% in colorectal carcinomas, and 14% in breast carcinomas [157,158,159].
III	Loss of an HLA class I locus	Found when both products of HLA-A, -B, or -C loci are coordinately downregulated [160,161].Since the levels of mRNA found in these tumor cell lines can frequently be upregulated in the presence of cytokines and low expression of transcription factors that bind to locus-specific DNA motifs can induce HLA-B locus downregulation, the assumption is that the mechanism of locus downregulation is often transcriptional [162].
IV	HLA class I allelic loss	This molecular defect has been reported in colorectal carcinoma LS411, with a chromosomal break point in the HLA-A11 allele [163], or in the cervical cell lines CC11 and CSCC7 [164] or 808 and 778 [165,166].
V	Compound phenotype	Requires a combination of at least two different alterations.Perea et al. recently reported a mechanism responsible for a total HLA class I loss in approximately 60% of studied small cell lung carcinoma samples. It is the combination of HLA haplotype loss together with a transcriptional downregulation of HLA-A, B and C genes [167].
VI	Failure to respond to interferon (IFN)	This altered phenotype is found when tumor cells express basal levels of HLA class I antigens, but they do not respond to the stimulation of HLA class I expression with different cytokines, such as α and γ interferons (IFNs). For instance, the renal cell carcinoma Caki-2 does not have DNA-binding activity for IFN regulatory factor-1 or signal transducer and activator of transcription (STAT-1) [168].
VII	Downregulation of classical HLA molecules (Ia) with aberrant expression of non-classical HLA molecules (Ib)	Based on unique HLA class I tissue distribution, that is used by the tumors to avoid both T and NK cell responses. It enables cancer cells to escape CTL responses by losing HLA-A, B, C. At the same time, these tumor cells, by engaging HLA-Ib molecules with NK inhibitory receptors, are resistant to NK lysis [156,169,170].

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
