# Peer review of "MHC Class I Regulation: The Origin Perspective"

_cancers, 2020, doi:10.3390/cancers12051155_

Round 1

Reviewer 1 Report

In their manuscript entitled "MHC class I expression in healthy and cancer state: the origin perspective" Sznarkowska et al. first introduce the so called "concept of viral origin of MHC" by Villarreal and in the second part focus on the regulation of MHC class I expression, which in part is brought about by non-coding RNA.

The split up of the manuscript in two almost separate conceptual parts is a major weakness: the first part is a strange melange of science, far-fetched hypotheses and more problematic statements. For instance, the appearance of MHC components does not appear to be such a singular event as the authors claim (lines 83-88). The immunoglobulin-like domain, lektin-like molecules, primitive histocompatibility systems and even RAG-like molecules have been identified not only in early vertebrates, but in other phylae as well (e.g. Sun et al. (1990) Science 250, 1729-1732; Fugmann et al. (2006) PNAS 103, 3728-3733; and many others), in sum suggesting the building blocks of MHC components have indeed evolved earlier. It is hard to envision how viruses should have "invented" the adaptive immune system rather than shuffling and modifying bits and pieces of their host's genes (which, nevertheless, would be an important contribution). In the view of this reviewer there is no stringent necessity to claim a non-Darwinian type of evolution of the MHC region by "agents able to understand and to extend an existing code" (deus ex machina?). The adaptive immune system may be a complex entity, but life in general is characterized by complexity and still generally considered to have evolved by an interplay of mutation and selection.

Considering the second part concerned with alterations in the expression of MHC I molecules, the whole virus origin evolution tale of MHC becomes rather dispensable. Here the authors focus exclusively on regulatory events brought about by various non-coding regions of DNA. Although interesting, the authors remain on a rather descriptive level and could have gone into much more mechanistic detail instead of stressing cloudy meaningfulness. Personally I miss an example for an RNA regulatory network, which is constantly put forward in interpretations by the authors, but obviously has not yet surfaced in real experimental results.

In summary, the combination of two little related parts in one review appears to be very problematic especially due to the highly speculative and strange first part. I seriously doubt the manuscript will appeal to a critical readership.

Author Response

Reviewer 1

We thank the reviewer very much for the valuable comments. They indicate that we did not succeed in the proper presenting the Luis Villareal concept of the viral-mediated evolution. This is a pity as his insight is extremly appealing to us. Therefore, in the revised version, we broadly discuss how viruses could have promoted the emergence of immunity (and other) networks and link it to the MHC class I regulation.

„The split up of the manuscript in two almost separate conceptual parts is a major weakness: the first part is a strange melange of science, far-fetched hypotheses and more problematic statements. For instance, the appearance of MHC components does not appear to be such a singular event as the authors claim (lines 83-88). The immunoglobulin-like domain, lektin-like molecules, primitive histocompatibility systems and even RAG-like molecules have been identified not only in early vertebrates, but in other phylae as well (e.g. Sun et al. (1990) Science 250, 1729-1732; Fugmann et al. (2006) PNAS 103, 3728-3733; and many others), in sum suggesting the building blocks of MHC components have indeed evolved earlier.

It is true that the separate components of the adaptive immunity or their equivalents have been identified in invertebrates and the jawless vertebrates as the reviewer points out. Still the complete and coherent network of adaptive immunity, with all its elements: immunoglobulins, TCRs, RAG and MHC, first appeared in jawed fish and it indeed was a sudden, big bang-like event.

Now we included in the main text the paragraph on the antigen receptors equivalents present in the jawless fish – an immediate ancestors of jawed vertebrates - and discuss how the occurence of separate components of adaptive immunity in this early vertebrates could have resulted from separate viral infections of diverse viral lineages (lines: )

„In the view of this reviewer there is no stringent necessity to claim a non-Darwinian type of evolution of the MHC region by "agents able to understand and to extend an existing code" (deus ex machina?). The adaptive immune system may be a complex entity, but life in general is characterized by complexity and still generally considered to have evolved by an interplay of mutation and selection.”

We have now dedicated the whole paragraph (3. Survival in the virosphere thanks to viral addiction modules) to present the conceptn of viral-mediated evolution.

„Considering the second part concerned with alterations in the expression of MHC I molecules, the whole virus origin evolution tale of MHC becomes rather dispensable. Here the authors focus exclusively on regulatory events brought about by various non-coding regions of DNA. Although interesting, the authors remain on a rather descriptive level and could have gone into much more mechanistic detail instead of stressing cloudy meaningfulness. Personally I miss an example for an RNA regulatory network, which is constantly put forward in interpretations by the authors, but obviously has not yet surfaced in real experimental results.”

We thank the Reviewer very much for this comment. In the revised version there is now paragraph: 6.Virus-mediated evolution and regulation of MHCI expression where we stress the role of retroviral elements in the control of interferon network [1] (lines 274-279). We also discuss the huge regulatory potential brought by long non-coding RNA HCP5 encoded within MHC region and containing a big chunk of endogenous retroviral sequence (HERV16). We hypothesise that the function of this RNA may have resulted from an ancient viral colonization which enhanced the host’s immunity network.

Reviewer 2 Report

In the current manuscript, Sznarkowska et al provide a thorough review of the theory that viruses have played a critical role in the evolution of human MHC gene organization and expression, as well as the alterations in MHC class I expression that have been observed in various human cancers and the impact of these alterations on immunotherapies. This manuscript is organized well and comprehensive.

I thought that the authors did a good job with their review. In my opinion
the manuscript provides a comprehensive look at the potential role viruses
played in the evolution of the MHC loci, and an aspect of the molecular
biology underlying the downregulation of MHC I in cancer cells. Besides
the grammatic errors I listed, upon rereading the manuscript it may
benefit the reader if the authors provide some additional text to ease the
transition between viruses and MHC polymorphisms and regulation of MHC I
during cancer (lines 188-189). For instance, describe the modulation that
has been observed with long noncoding RNAs and other immune pathways.

My comments are primarily grammatical:

line 87: replace with "big bang"

line 101: remove "another" or replace with "subsequent"

line 113: replace "add" with "adds"

line 272: remove the extra "at the"

line 292: remove "s" from "T cells"

line 374: replace "on PD-L1 status" with "of PD-L1 status"

line 406: "seems to be the way to go" sounds too colloquial; recommend replacing with "... in cancer appears critical for advancing our understanding of the course of this disease"

Author Response

We thank very much the Reviewer for the positive feedback and hope that the revised version of the manuscript will be satisfactory. We have siginificantly developed the viral origin perspective section and presented mechanisms used by viruses to aquire persistent relationship with the host, which also confers immunity. In the MHC regulation part, besides classic transcription factor-based regulation we have described the long non-coding RNA of retroviral origin (encoded within MHC region) which regulates various aspects of immune response and cell proliferation most probably via interactions with different regulatory RNAs.

Reviewer 3 Report

This review describes “the viral-mediated origin and evolution of the MHC class I region” and describes regulatory RNA networks involved in the regulation of MHC genes and their role in cancer.

Despite the fact that it is well accepted the role of virus in the evolution of human genome and the regulation of most cellular functions, including the immune response or antigen presentation, this review  is very speculative and there are not strong data supporting many of the information provided. For instance, the hypothesis that “viruses are likely the inventors of the immune system” is interesting, but there are no strong data supporting this concept. In fact, there are few data regarding the role of virus in the regulation of the expression of MHC molecules or in the regulation of the immune response. Instead, this review provides a lot of information with no clear connection with the topic such as those related to transcriptional regulation, methylation and histone acetylation, HLA class I altered phenotypes in cancer or immunotherapy. The authors have failed to show an evolutionary perspective in these sections and they have not shown a clear relation with a viral origin and what is the relevance of this origin.

From my point of view, this review is a mixture of different information with no clear connection among them and, in general, it is not convincing.

Author Response

We thank the Reviewer very much for the comments. In the revised version we have largely expanded the section on viral origin of adaptive immunity and dedicated paragraphes 1,2 and 3 to introduce the perspective of Luis Villareal on the role of viruses in evolution. We have also provided the emerging evidence validating this concept like the regulation of interferon response by endogenous retroviruses (ERVs) (lines:274-279) or the immune respose regulation by long non-coding RNA of retroviral origin (lines:280-305).

Round 2

Reviewer 1 Report

In their revised manuscript the authors the authors added details and explanations in the first part and streamlined the second part by focusing on the regulation of MHC I expression. By this the manuscript gained focus and became more legible for the reader. Still, the main point of criticism of one of the other referees and this reviewer has not yet been addressed in the revised manuscript: the gap between an evolutionary point of view on the MHC region and the potential role of viruses in it on one hand and the issues of MHC I regulation in tumors on the other. Here the manuscript should become much more detailed because, in the end, that is the main point of the manuscript. For example, the authors mention the role of the HPC5 element in the regulation of MHC class I molecules, however they omit to mention what is known in respect to cancer and MHC class I expression in cancers. A recent review in Cells on HCP5 (Kulsky, J.L., Cells 8, 480, 2019) may provide an example what a good review could provide. It would also be interesting to hear about other retroviral elements and their link to MHC class I expression, cancer biology and anti-cancer therapy. Furthermore, the authors should try to relate this type of information to the classification of MHC I expression defects in tumors, because otherwise this table would remain as an rather unlinked block of information in the ms.

The other problem I do still have with the manuscript is the exclusive point of view the authors take: the positive role of viral remnants in host DNA by design (even if they changed the wording to some extent in the revised ms). There is no question that these elements are there, abundant, and that they play roles in some regulation processes. However, what the authors claim to be there by "design" could have also come to existence by a process of co-evolution, a sort of arms race that in the end was won by the host who in some cases may have managed to exploit or at least productively integrate novel functions into its genome. By exclusively focusing on a speculative positive role of viruses the manuscript gets a kind of religious tenet that I feel is completely inappropriate for a review. The authors definitely need to adopt a more balanced view on the evolutionary aspects of retroviral elements.

Author Response

We thank the reviewer again for the constructive comments to the revised version of the manuscript. They let us realize which lines of arguments we have missed to comprehensively describe the impact of the persistent viral infections on the origin and evolution of adaptive immunity, the host evolution in general, and how it shows itself in the regulation (and deregulation) of MHC expression.

The comment of the reviewer:

Still, the main point of criticism of one of the other referees and this reviewer has not yet been addressed in the revised manuscript: the gap between an evolutionary point of view on the MHC region and the potential role of viruses in it on one hand and the issues of MHC I regulation in tumors on the other. Here the manuscript should become much more detailed because, in the end, that is the main point of the manuscript. For example, the authors mention the role of the HPC5 element in the regulation of MHC class I molecules, however they omit to mention what is known in respect to cancer and MHC class I expression in cancers. A recent review in Cells on HCP5 (Kulsky, J.L., Cells 8, 480, 2019) may provide an example what a good review could provide. It would also be interesting to hear about other retroviral elements and their link to MHC class I expression, cancer biology and anti-cancer therapy. Furthermore, the authors should try to relate this type of information to the classification of MHC I expression defects in tumors (…)

In the revised version we have extensively expanded the MHC regulation part brought by viral elements and created a new section on “MHC-regulating non-coding RNAs in cancer (chapter 8) where we provide information ncRNAs affecting MHC genes and their associations with cancer. The main points we added and/or expanded are:

  • ERVs elements are the binding sites for the immune transcription factors STAT1 and IRF1
  • ALUs and LINEs elements distribute miRNAs target sites to 3’UTRs of multiple transcripts
  • virus-derived elements from MHC region encode for multiple non coding RNAs
  • HCP5 oncogenic action and associations with cancer
  • virus elements-derived miRNAs targeting MHC genes and their association with cancer

The reviewer further says:

“The other problem I do still have with the manuscript is the exclusive point of view the authors take: the positive role of viral remnants in host DNA by design (even if they changed the wording to some extent in the revised ms). There is no question that these elements are there, abundant, and that they play roles in some regulation processes. However, what the authors claim to be there by "design" could have also come to existence by a process of co-evolution, a sort of arms race that in the end was won by the host who in some cases may have managed to exploit or at least productively integrate novel functions into its genome. By exclusively focusing on a speculative positive role of viruses the manuscript gets a kind of religious tenet that I feel is completely inappropriate for a review. The authors definitely need to adopt a more balanced view on the evolutionary aspects of retroviral elements”

Indeed in the manuscript we present the not widely appreciated view on the host evolution driven by persistent viruses. Though this concept was considered so important by The American Society for Microbiology that it has commissioned Villarreal to write a book on it (Villarreal, Viruses and the evolution of Life, 2005, ASM Press). We think it is important to bring this concept to cancer researchers as it might shed a light on complex immune system regulation and deregulation brought by viral elements and ncRNAs they encode. Now In the revised ms, particularly in the introduction, chapter 3 and 4, we discuss more extensively why the concept of Villarreal and Witzany, emphasizing co-operation and symbiosis, in our view explains the origin and evolution of the adaptive immunity (and any other regulatory networks) better than the individual based selection of errors.

Reviewer 3 Report

The revew is very speculative.

Author Response

The manuscript brings the concept of the viral origin of an adaptive immunity and, in the broader sense, the impact of persistent viruses on the host evolution It is not speculative, but on the contrary, deeply rooted in the experimental virology [1–8] and the philosophy of science [9–12]. It stems from the endosymbiotic theory of Lynn-Margulis [13] emphasizing the co-operation and symbiosis as a driving force of life, in contrast to the individual fittest type selection of errors (put forward as ‘selfish gene’ by Dawkins).

We think it is important to bring this concept to cancer researchers as it might shed a light on complex immune system regulation and deregulation in cancer, brought by viral elements and regulatory ncRNAs they encode.

  1. Villarreal, L.P. Viruses and the evolution of life; ASM Press, 2005; ISBN 1555813097.
  2. Villarreal, L.P. Origin of group identity : viruses, addiction, and cooperation; Springer, 2009; ISBN 9780387779973.
  3. Villarreal, L.P. Persistence pays: how viruses promote host group survival. Curr. Opin. Microbiol. 2009, 12, 467–472.
  4. Villarreal, L.P. The source of self: Genetic parasites and the origin of adaptive immunity. Ann. N. Y. Acad. Sci. 2009, 1178, 194–232.
  5. Villarreal, L.P. Viral ancestors of antiviral systems. Viruses 2011, 3, 1933–1958.
  6. Villarreal, L.P.; Witzany, G. Viruses are essential agents within the roots and stem of the tree of life. J. Theor. Biol. 2010, 262, 698–710.
  7. Moelling, K. What contemporary viruses tell us about evolution: A personal view. Arch. Virol. 2013, 158, 1833–1848.
  8. Moelling, K.; Broecker, F. Viruses and evolution - Viruses first? A personal perspective. Front. Microbiol. 2019, 10.
  9. Witzany, G. Natural genome-editing competences of viruses. Acta Biotheor. 2006, 54, 235–253.
  10. Witzany, G. Two genetic codes: Repetitive syntax for active non-coding RNAs; non-repetitive syntax for the DNA archives. Commun. Integr. Biol. 2017, 10, e1297352.
  11. Villarreal, L.P.; Witzany, G. The DNA habitat and its RNA inhabitants: At the dawn of RNA sociology. Genomics Insights 2013, 6, 1–12.
  12. Villarreal, L.P.; Witzany, G. That is life: communicating RNA networks from viruses and cells in continuous interaction. Ann. N. Y. Acad. Sci. 2019, 1447, 5–20.
  13. Sagan, L. On the origin of mitosing cells. J. Theor. Biol. 1967, 14, 225-IN6.

Round 3

Reviewer 1 Report

The authors present an improved manuscript and have tried to respond to the criticism. It is a pity that they still stick so religiously to the ideas of Villarreal and almost ignore alternative explanations. I would therefore suggest to publish the manuscript not as a review (because of a lack of balance) but as an "opinion" or "hypothesis" type of article.

Author Response

We thank the reviewer for the comments and the suggestion on changing the manuscript type. We do agree that the manuscript is better suited for the 'perspective' article as it does present a perspective on the adaptive immunity origin and seeks for its validation in the MHC regulation network.